# Self-Luminous Wood Coatings with Carbon Dots/TiO_2_ Grafted Afterglow SrAl_2_O_4_: Eu, Dy Core-Shell Phosphors for Long-Lasting Formaldehyde Removal

**DOI:** 10.3390/polym15092077

**Published:** 2023-04-27

**Authors:** Longfei Zhang, Ying Wang, Limin Peng, Zhilin Chen, Shaoyi Lyu, Siqun Wang

**Affiliations:** 1Research Institute of Wood Industry, Chinese Academy of Forestry, Beijing 100091, China; zhanglongfei@caf.ac.cn (L.Z.);; 2Co-Innovation Center of Efficient Processing and Utilization of Forest Resources, Nanjing Forestry University, Nanjing 210037, China; 3National Forestry and Grassland Administration Key Laboratory of Plant Fiber Functional Materials, Fuzhou 350002, China; 4Center for Renewable Carbon, University of Tennessee, Knoxville, TN 37996, USA

**Keywords:** carbon dots, core-shell microspheres, SrAl_2_O_4_: Eu^2+^, Dy^3+^ phosphor, titanium dioxide, surface modification, photocatalytic degradation

## Abstract

Long-term relief of indoor volatile pollution has become a competitive issue worldwide in both visible and dark environments. A novel self-luminous wood coating with carbon dots (CDs)/titanium dioxide (TiO_2_) nanomaterial coated SrAl_2_O_4_: Eu^2+^, Dy^3+^ (CDs/TiO_2_@SAO) composite was prepared for the long-term degradation of formaldehyde through a simple sol-gel method. The microstructure, chemical composition, ultraviolet-visible (UV-vis) spectra, and long-lasting fluorescence of the CDs/TiO_2_@SAO photocatalyst were analyzed to illustrate the mechanism for degrading formaldehyde. The obtained CDs with a particle size of ~2–7 nm have a good graphite structure and presented good absorption in visible light. In addition, owing to the synergistic effect of the CDs/TiO_2_ nanomaterial coating layer and the long-afterglow luminescence of the SAO phosphor, the CDs/TiO_2_@SAO composite can absorb a part of the visible light for photocatalytic degradation and store luminous energy efficiently at daytime so as to give out visible luminescence continuously for a few hours in the darkness. Furthermore, the functional wood coatings with CDs/TiO_2_@SAO composite presented continuous and efficient photocatalytic activity in the presence and absence of light exposure. The current research could provide a new strategy for designing an efficient photocatalyst for degrading formaldehyde pollution in the daytime with a visible light supply and in an indoor dark environment without an external light source.

## 1. Introduction

Recently, indoor volatile organic pollutants (VOCs) have attracted more attention from researchers [1,2,3]. It is well known that construction materials and indoor decorations such as paintings, wallpapers, textiles, and wood functional panels generally release formaldehyde and other organic volatiles [4,5]. Among these indoor organic pollutants, the formaldehyde hazardous volatile pollutant is regarded as a typical air pollution owing to its potential threat to human health. Generally, the highest allowable concentration of formaldehyde in indoor air should be limited to less than 0.08 mg/m^3^ [6,7]. People in the living environment will have an obvious discomfort in mentality and physiology once the formaldehyde concentration reaches more than 0.10 mg/m^3^. Numerous studies have indicated that long-term exposure to the atmosphere containing a high concentration of formaldehyde could induce respiratory diseases, chronic poisoning, and even death [8,9]. Therefore, it is of great significance to develop novel approaches and materials for removing formaldehyde from our living environment.

Among the technologies developed for the treatment of organic pollutants, the regular strategies include plant purification, activated carbon adsorption, photocatalysis, and high-energy plasma treatment [10,11,12]. In such processes of controlling the harmful formaldehyde volatile, photocatalytic oxidation has been considered a green and economical technology owing to its simple process, low energy consumption, and thorough degradation of pollutant [13]. For the numerous catalytic materials, anatase titanium dioxide (TiO_2_) was regarded as the most effective light-induced catalytic material owing to its high cost-performance ratio, excellent chemical stability, and good ultraviolet (UV) resistance. However, traditional anatase TiO_2_ catalysts generally need to initiate under high-energy UV lamps or limited UV light in the sunlight. The visible light in daylight cannot play an effective catalytic activity [14]. Additionally, pristine anatase TiO_2_ cannot degrade indoor formaldehyde in the darkness, which limits its practical application. Considering the rich indoor visible light in the daytime and long hours for people sleeping at night, it is necessary to modify the anatase TiO_2_ to widen its photo-responsive wavelength, thereby resulting in a good catalyst with a visible light and darkness response for degrading formaldehyde. 

The photocatalytic decomposition of formaldehyde volatiles has been difficult because of the common deactivation of modified anatase TiO_2_ composite, especially in visible and dark environments. Luminescent ions and nanomaterial doping processes were vital methods for the modification of long-lasting phosphors [15]. Various strategies (metal deposition, graphite carbon nitride modification, and phosphor decoration) have been developed for the visible-light modification of TiO_2_-based photocatalysts [16,17,18]. However, the most traditional graphite carbon nitride and metal dots are expensive or contain toxic metal-ions, which are harmful to human health in indoor applications. Carbon dots (CDs), as a classic fluorescent nanomaterial, have gained increasing attention for bio-imaging, sensing, and catalytic fields because of their low toxicity, wide absorption spectrum, good electronic conductivity, and doped-dependent photoluminescence (PL) behaviors [19,20,21]. Till now, hydrothermal synthesis of CDs from biomass (wood, grass, and bamboo particles) has been proven to be a low-cost and effective strategy, thereby producing doped CDs with diverse absorption in visible light regions [22]. The fluorescent CDs within 10 nm in size generally have small band gaps (≤2.0 eV) owing to their nanoscale sizes and quantum effects, which presents great potential in catalysis fields [23]. For the CDs/TiO_2_ composite photocatalyst, the introduction of CDs is conductive and enhances the photo-induced electron transfer of the TiO_2_ system. Gao et al. [24] synthesized a catalyst with lignin-based CDs and TiO_2_ sheets, which enhanced both the adsorption capacity and activity of the carbon dioxide. Martins et al. [25] applied nitrogen (N) doped CDs to prepare composite catalysts, which exhibited good degrading activity of NOx under visible light irradiation. However, not all CDs can contribute to visible-light catalytic activity. CDs with N- and phosphorus (P)-doping can help the adsorption of molecular oxygen and offer the high efficiency of CDs/TiO_2_ composites for decreasing organic pollution [26].

Moreover, it is crucial to prolong the lifetime of the composite catalyst system and further enhance its comprehensive activity in photocatalysis. SrAl_2_O_4_: Eu^2+^, Dy^3+^ (SAO) is a typical rare-earth strontium aluminate long-afterglow self-luminous phosphor, which is beneficial for energy harvesting in the daylight (UV or visible light) and emitting long-lasting visible light under dark conditions. It is widely applied in our lives, such as in indication marking, luminescence sensing, luminous plastic, the ceramic industry, and photocatalytic fields [27,28,29,30]. The composites with SAO phosphors generally store light energy during the daytime and emit visible light for minutes to hours in the dark environment. However, SAO phosphors tend to cause hydrolysis under humid or watery environments, which accelerates the crystalline destruction of SAO particles and reduces their self-luminous activity. Moreover, the band gaps of SAO phosphors are obviously higher than those of typical CDs and TiO_2_ photocatalysts [31]. In our previous report, we showed that a sol-gel-induced SiO_2_-coating layer on SAO phosphors can prevent the degradation of O-Sr-O bonds even in an aqueous condition [32]. Therefore, a buffer binding layer modification is necessary to improve the water resistance and adhesion with other catalytic nanoparticles.

To our knowledge, there are no reports in the literature concerning the use of CDs/TiO_2_/SAO composites in the photocatalytic removal of formaldehyde and other indoor organic volatiles. Moreover, studies of the oxidation of formaldehyde using a photocatalyst with long-lasing visible light activity are still scarce. The objective of this research was to develop a novel self-luminous wood coating with carbon dots (CDs)/titanium dioxide (TiO_2_) nanomaterial coated SrAl_2_O_4_: Eu^2+^, Dy^3+^ (CDs/TiO_2_@SAO) composite for the long-term degradation of formaldehyde. In this work, a novel strategy for the reasonable utilization of CDs from wood sanding dust and the doped composite catalyst on wood surfaces for the degradation of formaldehyde was explored. Fluorescent CDs were first prepared by the hydrothermal process of wood sanding dust and then sol-coated with TiO_2_ and SAO phosphor to obtain a core-shell CDs/TiO_2_@SAO composite coating on the wood. The photo-oxidation effect of formaldehyde under UV irradiation, visible light, and a dark environment was demonstrated. The potential mechanism for the enhanced photocatalytic activity of CDs/TiO_2_@SAO composite coating has been further discussed.

## 2. Materials and Methods

### 2.1. Materials

Anatase TiO_2_ powder with a 25 nm primary particle size, tetraethyl orthosilicate (TEOS, 98%), citric acid monohydrate (CA), ethanol (C_2_H_5_OH), and pH buffer solutions with a pH of 3.0 were purchased from Aladdin. SAO phosphors with an average particle size of 6.2 μm were obtained by Lu Ming Luminescent Materials Co., Ltd. (Liaoning Province, China). Wood sanding dust waste (moisture content was about 4.6%) and the UV-initiated aqueous curing paint were supplied by Ruixin Decoration Co., Ltd. (Zhejiang Province, China). Tris (hydroxymethyl) phosphine oxide (THPO, 85%) was purchased from Shanghai Gaoming Co., Ltd. (China). The fir wood (Cunninghamia lanceolata (Lamb.) Hook.) samples were supplied by the Chinese Academy of Forestry (Beijing, China). Deionized water was used to extract the wood sanding dust and prepare solutions for all the experiments. 

### 2.2. Synthesis of N,P-Doped CDs from Recycling Wood Sanding Dust

The N,P-doped CDs were synthesized from recycled sanding dust via a two-step hydrothermal process. Shortly, sanding dust and 150 g of deionized water were mixed with a solid-to-liquid ratio of 1/12.5. The mixture was then transferred to autoclaves (250 mL) in an oven (165 °C) and pre-extracted for 1.0 h. The brownish pre-extraction (160 g) was filtered with a Syringe Filter (0.45 μm) and then transferred to a 250 mL sealed reactor with 8 g of THPO. The reactor was then processed at 220 °C for 6 h for carbonization. The resulting solution was cooled down, centrifuged at 8000 rpm for 8 min, and vacuum freeze dried to obtain N,P-doped CDs. For comparison, the CD materials without P-doping were also synthesized using the sanding dust directly in the same process, denoted as NCDs.

### 2.3. Formation of CDs/TiO_2_@SAO Composite Photocatalyst

The modified CDs/TiO_2_@SAO composite photocatalyst was obtained via a silica sol-gel method. Specifically, TEOS (0.8 mL) was mixed with ethanol (9 mL) and pH 3.0 buffer (2 mL) in a 65 °C water bath for 2 h. Uniform pre-dispersion of CDs/TiO_2_ was prepared by adding 0.05 g of TiO_2_ and 0.1 mg of N,P-doped CDs into ethanol (20 mL) and buffer (3 mL) solutions. Furthermore, CDs/TiO_2_ dispersions were transferred to the TEOS sol and vigorously stirred for 45 min at 65°C. Subsequently, SAO phosphors (0.8 g) were added and continually stirred for 15 min, in order to obtain uniform gel-coated phosphors. Finally, the obtained particles were washed using an alcohol solution until the upper centrifugal liquid was colorless under a 365 nm UV-lamp. The cleaned precipitation was then dried at 50 °C for 24 h to obtain the CDs/TiO_2_ modified SAO core-shell phosphors, denoted as CDs/TiO_2_@SAO. 

### 2.4. Formation of Photocatalytic Wood Coatings with CDs/TiO_2_@SAO Composite

The pre-weighed phosphors (3.6 g) were added to a water-based photo-curing paint (50 g) with mechanical stirring for 15 min to obtain a uniform suspension. The self-luminous suspension was then sprayed on a piece of wood (100 × 45 × 5 mm^3^) using a spray gun with a nozzle diameter of 0.5 mm under an air pressure of ~0.7 MPa. Subsequently, the samples after spraying treatment were processed using a UV-curing device to obtain photocatalytic wood materials. The coating thickness was controlled within 80–100 μm to maintain the apparent color of wood. For comparison, other modified wood materials with the pristine TiO_2_ and SAO coatings were prepared using similar processes. Finally, all the wood samples in the absence and presence of catalytic coating were placed in a room (20 °C; 55% relative humidity) for further tests. Figure 1 shows the preparation process of the CDs/TiO_2_@SAO core-shell composite and the resulting wood materials with long-term photocatalytic coatings.

### 2.5. Material Characterization

Morphologies of the dispersed CDs and modified SAO phosphors were observed using a FEI Tecnai G2-F30 transmission electron microscope (TEM; Hillsboro, USA). A drop of highly diluted CDs aqueous solution or phosphor dispersion was placed onto the cooper grid with a holey amorphous carbon film and left to dry in the air until being used for the TEM microscope. High-resolution TEM (HRTEM) images were used to analyze the lattice fringes of CDs. The surface topographies of CDs in three-dimensional (3D) and two-dimensional (2D) modes were performed using a Bruker atomic force microscope (AFM; Karlsruher, Germany). Fourier-transform infrared (FTIR) spectra of the solid samples were recorded using a Nicolet IS10 spectrometer (Massachusetts, USA). Thermogravimetric analysis (TGA) curves of the samples were obtained using a Netzsch TG STA 449 F5/F3 analyzer (Bavaria, Germany) in nitrogen atmosphere using a heating rate of 10 °C/min up to 800 °C. A Hitachi 8020 scanning electron microscope (SEM; Tokyo, Japan) equipped with an energy-dispersive spectroscopy (EDS) detector was used to observe the morphologies of the phosphors and wood coatings and to evaluate the distribution of catalyst using the element-mapping mode with the same scanning time. Powder X-ray diffraction (XRD) patterns were recorded using a D8 XRD meter (Brucker, Germany) with a scan rate of 8°/min. PL spectra of the CDs and phosphor samples were recorded using an FLS100/FS5 spectrometer (Edinburgh, UK). The same quality of phosphors in the presence and absence of CDs/TiO_2_ coatings was used for the comparative analysis of the PL emission. The Commission Internationale de L’Eclairage (CIE) coordinates of luminescent samples are calculated using the PL spectra. The PL decay status and afterglow lifetime were monitored in the spectrometer (excitation illumination: 1000 lx, excitation time: 15 min). The emission spectra of the indoor compact fluorescent light (CFL) and the sunlight (116°20′ E, 39°56′ N) at daytime were obtained using an S-5000 spectrometer (Hamamatsu, Japan). The absorbance spectra of CDs solutions were measured using a GD-54T UV-visible (UV-vis) spectrometer (Shanghai, China). The UV-vis diffuse reflectance spectra of photocatalyst were performed using a Shimatsu UV-3600 spectrometer (Tokyo, Japan). 

### 2.6. Photocatalytic Activities Characterization

Volatile formaldehyde represents a major organic pollutant of VOCs in indoor environments, and here it was chosen as a representative to evaluate the photocatalytic behaviors. All the photocatalytic experiments were carried out in a closed reactor with a volume of ~100 L to simulate the dynamic degradation behaviors of volatile formaldehyde pollutants. Three different light source conditions, such as a 365 nm UV lamp, visible light from a compact fluorescent light (CFL, 8 W), and a dark environment, were chosen to analyze the catalytic efficiency of functional wood materials. The schematic of the reactor for formaldehyde degradation in the presence and absence of a light source is shown in Figure 2. Prior to the experiment, the reactor was placed in a room (20 °C; 55% relative humidity) under the darkness for more than 24 h. Subsequently, the wood materials were put on a suspended wire mesh with a total area of ~135 cm^2^, and the required quantity of formaldehyde was injected into the closed reactor. Typically, the initial concentration of volatile formaldehyde in the reactor was kept at ~0.2 mg/m^3^ for all the experiments. When the formaldehyde concentration reached a dark-adsorption equilibrium, photocatalysis experiment was started by turning on the UV lamp or visible light. 

Specifically, for the long afterglow photocatalytic experiment in the darkness, the wood samples with CDs/TiO_2_@SAO composite coating were pre-exposed to CFL visible light for 45-min before the formaldehyde injection. The degradation efficiency of formaldehyde was calculated by (*C*_0_ − *C_t_*)/*C*_0_, where *C*_0_ is the reactant concentration after adsorption equilibrium and *C_t_* is the reactant concentration after irradiation from light or preservation in darkness for a certain time. For comparison, the wood materials with pristine TiO_2_ and SAO composite coatings were also treated in the same process for evaluation of the degradation efficiency owing to the CDs and SAO modifications.

## 3. Results and Discussions

### 3.1. Characterization of Fluorescent N,P-Doped CDs

The TEM and HRTEM characterization results of N,P-doped CDs (Figure 3a,b) show that the as-prepared CDs are uniform in size and well dispersed. The particle size distributions of N,P-doped CDs were mainly in the range of 2–7 nm with an average size of 4.5 nm (100 random nanoparticles were accounted for CDs) (Figure 3c). HRTEM images of the N,P-doped CDs (Figure 3b) reveal that most of the CDs are crystalline with well-resolved lattice fringes. The inset lattice spacing in the HRTEM image was ~0.21 nm, indicating that the spacing array can be assigned to the crystalline phase (1120) of graphite, which is consistent with our previous report on biomass-derived CDs [20]. The AFM images (Figure 3d,e) also indicate the formation of well-dispersed particles in spherical shape. Moreover, the hyperfine nanoparticle deposition depth of CDs dispersion was mainly located in the range of 6.5–8.2 nm (Figure 3f), which indicates the nanoscale roughness was deposited owing to 1–4 layers of well-dispersed CDs.

The chemical groups of the N,P-doped CDs were further analyzed using FTIR (Figure 3g). The FTIR spectra reveal that the N,P-doped CDs were highly functionalized with oxygen-containing groups. The broad peak around 3490–3050 cm^−1^ is ascribed to the stretching vibrations of the O–H or N–H bonds, indicating the abundant existence of hydroxyl and carboxyl groups in the CDs [33]. The shoulder peak at 2910 cm^−1^ is attributed to various C−H vibrations. The three sharp peaks at 1622, 1495, and 1176 cm^−1^ are attributed to the vibration modes of N–H, COOH, and P−O−C, respectively [34]. Other peaks corresponding to P−OH (885 cm^−1^) and P−C (762 cm^−1^) have also been observed. These results imply that abundant amino, carboxyl, and phosphate groups were present on the surface of N,P-doped CDs. The EDS diagram and elemental contents of N,P-doped CDs further indicate that the CDs contain C, O, N, and P elements at concentrations of 44.05, 39.18, 2.31, and 14.46 wt%, respectively (Figure 3h). Thus, these results indicate that the N,P-doped CDs are mainly composed of carbon-core and active surface groups owing to their intrinsic C, O, N, and P elements.

### 3.2. Characterization of CDs/TiO_2_@SAO Core-Shell Composite Photocatalyst

A novel coated phosphor of long-lasting afterglow SAO decorated with CDs/TiO_2_ nanoparticles was prepared using a simple sol-gel coating method. Figure 4a shows the typical crystalline structure of dispersed TiO_2_ in a rectangular shape. Prior to the coated modification of SAO, the microstructure of CDs/TiO_2_ composites was evaluated (Figure 4b). It is clear that apparent agglomerations occurred between TiO_2_ and CDs nanoparticles due to the abundant hydroxyl functional groups on the CDs and TiO_2_ nanoparticles, which is beneficial to the further modification of CDs/TiO_2_ composites on the surface of SAO phosphor via a sol-gel process. Figure 4c shows the TEM image of CDs/TiO_2_@SAO phosphor. It clearly illustrates the successful coupling of nano-scaled coating onto the SAO particles.

The HRTEM of CDs/TiO_2_ composites further reveals that smaller spherical particles of CDs intermixed with larger rectangular crystals of TiO_2_ were clearly identified due to the different lattice fringes, as shown in Figure 4d. The clear lattice fringe of 0.35 nm was assigned to the (101) diffraction plane of the anatase phase of TiO_2_ particles, while the fringe of 0.21 nm matched that of CDs [35]. These results fully proved that both CDs and TiO_2_ nanoparticles were successfully coated on the surface of SAO phosphor after the sol-gel strategy. The SEM images of uncoated SAO and CDs/TiO_2_-coated SAO phosphors are shown in Figure 4e,f. It was apparent that the surface of coated SAO particles became rougher than that of uncoated SAO, which further illustrates that a protective coating layer was formed on the SAO particles. All these results further confirm that the fluorescent CDs and photocatalytic TiO_2_ nanoparticles were uniformly coated on the afterglow SAO phosphors.

The structure and thermal decomposition of the SAO phosphors in the absence and presence of CDs/TiO_2_ coating layer were further identified by the XRD and TGA analyses. Figure 5a exhibits the XRD patterns for pristine TiO_2_ and the CDs/TiO_2_ nanoparticles coated on the SAO phosphors. Both the anatase TiO_2_ and SAO particles were crystalline with typical characteristic peaks. Compared with TiO_2_, the XRD pattern of CDs/TiO_2_ after CDs modification exhibits similar peaks, whereas the intensity for CDs/TiO_2_ shows a slight increase at 20–30° compared with the baseline. This observation is consistent with our previous result that hydrothermal biomass-based CDs were primarily amorphous structures with wide diffraction [20]. For the coated CDs/TiO_2_@SAO photocatalyst, the characteristic peaks at (011), (211), and (031) for SAO weakened with additional CDs and TiO_2_ coatings were presented [32]. A new little peak assigned to the (101) crystal of TiO_2_ at 25.4° also appeared, whereas other characteristic peaks of TiO_2_ were not obvious owing to their overlapping with the strong peaks of SAO phosphors. This confirms that the CDs/TiO_2_ coating on SAO particles has no adverse effect on the typical crystalline structure of TiO_2_ and SAO phosphors.

The TGA and DTG results of modified phosphors are shown in Figure 5b,c. The TGA curves of unmodified SAO and TiO_2_ showed a rapid mass loss at 40–150 °C and then remained constant up to 800 °C. It is mainly attributed to the occurrence of adsorbed water owing to the hydroxyl groups on the microspheres and nanoparticles. For the CDs/TiO_2_@SAO composite photocatalyst, the decomposition process presents three stages. The steep decomposition peak centered at 150–255 °C was ascribed to the elimination of surface groups on CDs; this is consistent with the pyrolysis result of biomass-based CDs [20]. As the temperature continues to rise from 250 °C to 550 °C, the continuous mass loss (~8.6%) is consistent with the fluctuating pyrolysis of silane-containing polymers physically bonded to the surface of SAO particles [28]. The inset in Figure 5c is the photographs of CDs/TiO_2_@SAO at daylight and dark conditions after pyrolysis. The strong green emission of photocatalyst at night illustrates that the self-luminous activity of CDs/TiO_2_@SAO can be maintained even after 800 °C thermal treatment.

The typical emission spectra of the indoor CFL light and the sunlight at daytime are shown in Figure 5d. It is clear that sunlight is composed of strong UV light (315–400 nm), visible light (400–770 nm), and weak near-infrared light (770–900 nm). Compared the sunlight, the spectrum of indoor CFL is in the range of visible light at 400–700 nm, including blue light (400–485 nm), blue-green light (485–510 nm), green light (510–565 nm), and other wavelengths of light. Therefore, the key to improving the photocatalytic activity of wood coatings is to broaden the response of the photocatalyst in an indoor visible light environment. The optical properties of the samples were analyzed with the UV-vis spectra. The UV-vis absorption spectra of CDs in the presence and absence of P-doping are shown in Figure 5e. The maximum absorption peak of the N,P-doped CDs was observed at 340 nm, which is attributed to the n − π* transitions of surface C=O and C−N bonds [36]. Whereas the shoulder at 265 nm is represented by the π − π* transition of the aromatic C=C bond from the carbon core. Compared with the NCDs without P-doping, the UV-vis spectrum of N,P-doped CDs also exhibits broad tailing at 370–475 nm, which helps absorb additional visible light owing to the oxygen-related P-doping groups and surface defects in the N,P-doped CDs [37]. Figure 5f shows the UV-vis diffuse reflectance of pristine TiO_2_, SAO phosphor, and CDs/TiO_2_@SAO composite photocatalyst. It can be seen clearly that the CDs/TiO_2_@SAO composite has a wider absorption band in the wavelength range of 400–700 nm, whereas the pristine TiO_2_ only exhibits a strong UV optical absorption edge at 400 nm corresponding to the wide band gap (~3.2 eV) as referenced by other researchers [38]. It is mainly due to the strong absorption of N,P-doped CDs and SAO phosphor in the UV-vis region, which may change the charge transfer process between the CDs and TiO_2_ in the coupling compound materials. Therefore, the optical absorption of the TiO_2_ system was extended to the UV-vis region by incorporation of the N,P-doped CDs and the self-luminous SAO phosphor, thus enhancing the photocatalytic degradation efficiency of the CDs/TiO_2_@SAO hybrid system for formaldehyde pollutant.

The PL emission spectra of N,P-doped CDs aqueous dispersion at different excitation wavelengths (340–490 nm) are displayed in Figure 6a. The PL intensity distinctly increased as the excitation wavelength increased from 340 nm to 430 nm and subsequently decreased with the increasing excitation wavelengths up to 490 nm. The maximum PL emission of the N,P-doped CDs appeared at λ_ex_ of 430 nm (blue), which is different from our previous biomass CDs without P-doping [20]. It was expected to expand the visible light absorption of CDs and enhance the photocatalytic activity of the CDs composite under visible lights. Moreover, the CIE chromaticity diagram of the N,P-doped CDs solution is present in Figure 6b. Especially, the PL emission of the SAO phosphor presented absolute green light with the CIE coordinates of 0.226 (x) and 0.469 (y), whereas the N,P-doped CDs depicted multicolor behavior tuned from 455 nm (blue) to 570 nm (yellowish-green). The tunable emission result of CDs is mainly attributed to the quantum effects and electron transitions between the σ and π molecular orbitals owing to the N,P-doping modification, which is consistent with our previous N-doping biomass-based CDs [20].

The excitation spectra of SAO phosphor (detected at 510 nm) and PL emission spectra (excited at 350 nm) under a dark environment of modified SAO phosphors are shown in Figure 6c. The excitation spectrum of SAO phosphor presented a continuous wide band centered at 360–428 nm, indicating that the SAO phosphors were effectively excited by UV light to visible light. It is attributed to the typical 4f^6^5d^1^ to 4f^7^-transition of Eu^2+^ ions in SAO crystals [39]. Furthermore, the PL emission intensity of CDs/TiO_2_@SAO modified photocatalyst was ~5% lower than that of the SAO sample, while the peak for the SAO phosphors before and after modification was unchanged. It can be attributed to the slight light scattering activity due to the CDs and TiO_2_ nanoparticles on the core-shell SAO surface owing to the sol-gel modification. The long afterglow luminescence decay curves of CDs/TiO_2_@SAO and SAO phosphors are shown in Figure 6d. After removal of excitation irradiation, the two phosphors presented a strong initial brightness up to 1635 and 1855 mcd/m^2^ in the dark environment. Subsequently, a rapid decay was occurred during the first 2 min and then a slow decay behavior up to 3600 s. The brightness of the CDs/TiO_2_@SAO and SAO phosphors at 3600 s (6 h) was still 3.19 mcd/m^2^ and 3.55 mcd/m^2^, respectively. These values were obviously higher than the recognition minimum limit of 0.32 mcd/m^2^ by the naked eye at night. The afterglow luminescence of CDs/TiO_2_@SAO was not obviously changed after the CDs and TiO_2_ coating, indicating that the sol-gel CDs/TiO_2_ coating layer was suitable for the modification of SAO phosphors. Therefore, these core-shell microspheres with self-luminous SAO phosphors and CDs/TiO_2_ photocatalysts have a vast potential application in photocatalytic degradation under indoor visible light and dark environment.

### 3.3. Morphologies of Wood Coatings with CDs/TiO_2_@SAO Composite Photocatalyst

The composite photocatalyst was further introduced to the typical UV-curing paint matrix to prepare wood coatings. Figure 7 shows the surface microstructure of the control wood and functional wood materials after the UV-curing paint coating and CDs/TiO_2_@SAO coating. For the control wood sample without surface coating modification (Figure 7a,d), it was clear that the wood shows a porous surface with a large number of tracheid lumens, wood rays, and cracks in micrometer scale, which is beneficial for the filling and adhesion of the paint substrate with the CDs/TiO_2_@SAO particles. The paint coatings were observed clearly on the wood samples (Figure 7b,e), and covered the original surface morphology of the porous wood skeleton to form a relatively smooth film. This can be explained by the fact that the paint polymers adhere to the wood surface owing to their strong binding effect. Additionally, the composite paint coatings with CDs/TiO_2_@SAO photocatalytic particles were attached tightly to the wood (Figure 7c), suggesting a good filling and adhesion of the coatings to the porous wood skeleton. The SEM image of the CDs/TiO_2_@SAO composite photocatalytic coatings in higher magnification indicated that random particles were anchored tightly by the film-forming paint (Figure 7f). To further determine the distribution of the CDs/TiO_2_@SAO photocatalyst on the coatings, the EDS elements mapping of the typical elements for SAO, TiO_2_, and CDs were examined (Figure 7g–j). The EDS elements mapping clearly showed the existence of Sr, Si, C, and N in the photocatalytic coatings, and all these elements were distributed uniformly in the composite coatings. These EDS results were in line with the results obtained from the SEM of coatings. The uniform photocatalytic coatings on wood substrate is essential for the practical indoor application in capturing and eliminating the volatile formaldehyde pollution.

### 3.4. Photocatalytic Activity and Mechanism of Formaldehyde Reduction

Volatile formaldehyde represents a major organic pollutant in indoor environments, and here it was chosen as a representative to verify the degradation behaviors of the functional wood materials with CDs/TiO_2_@SAO composite coatings. Firstly, the wood samples were put into the reactor in darkness for 24 h to eliminate the long-lasing afterglow fluorescence effect. Prior to irradiation, quantitative formaldehyde pollution was injected into the reactor ~30-min in advance to ensure the adsorption-desorption equilibrium. Figure 8a presents the formaldehyde degradation curves of the wood materials with CDs/TiO_2_@SAO composite coatings under continuous exposure to visible light (λ ≥ 400 nm). As shown in Figure 8a, the blank experiment presents that the self-degradation activity of the wood with pristine SAO and TiO_2_ coatings can be ignored owing to a slight decrease even after 160-min irradiation with visible light, which is mainly contributed to the slight adsorption effect owing to the porous structure of wood materials. The wood with CDs/TiO_2_ and CDs/TiO_2_@SAO composite coatings shows higher photocatalytic activities than that of the pristine TiO_2_ and SAO coatings under the visible light condition. It is probably due to the enhancement of electro-hole separation efficiencies in the CDs/TiO_2_ and CDs/TiO_2_@SAO composites [40,41]. For the CDs/TiO_2_@SAO composite coatings, the degradation reaction of formaldehyde reached ~60% in a 160-min cycle, which implies that the CDs/TiO_2_@SAO composite is a promising candidate for indoor volatile pollutants. 

Figure 8b shows the formaldehyde degradation curves of the wood materials with CDs/TiO_2_@SAO composite coatings under intermittent UV or visible light exposure. The photocatalytic process of formaldehyde was divided into two parts, such as photo degradation (keep lighting for 20 min) and darkness degradation (keep in the dark for 40 min). It can be seen clearly that the photocatalytic response was obviously improved with the additional energy supply (UV or visible light) for the degradation system owing to the good energy storage-release behavior of SAO phosphors. This is consistent with the long-lasting afterglow decay of CDs/TiO_2_@SAO composites (Figure 6d). Additionally, the photocatalytic degradation efficiency under UV irradiation was higher than that under visible illumination, which strongly agrees with the variation tendency in the UV-vis diffuse reflectance spectrum of the CDs/TiO_2_@SAO composite (Figure 5f).

Owing to the afterglow fluorescence decay and potential photocatalytic activity of the CDs/TiO_2_@SAO composite coatings in the darkness, the long-lasting self-luminous degradation of formaldehyde after an initial light exposure was performed to simulate the catalytic activity of wood materials at night when the visible light is turned off (Figure 8c). For comparison, the photocatalytic behaviors of TiO_2_ and CDs/TiO_2_@SAO composites have also been examined in the same experimental conditions. As shown in Figure 8c, when the external CFL visible excitation light (keep lighting for 60 min) was turned off, the TiO_2_ coated wood materials showed negligible photocatalytic activity for the degradation of formaldehyde. However, the functional wood with CDs/TiO_2_@SAO composite coatings still maintained photocatalytic activities at a certain level for ~360 min owing to the self-provided light from the SAO self-luminous phosphors. Though the afterglow decay lifetime of the wood with CDs/TiO_2_@SAO composite coatings was obviously more than 6 h (≥3.19 mcd/m^2^), the degradation activity after 180 min seemed negligible owing to the relatively weak inner-lighting of the CDs/TiO_2_@SAO photocatalytic system. Figure 8d shows the recycling stability of the functional wood with CDs/TiO_2_@SAO composite coatings for the degradation of formaldehyde. The photocatalytic recyclability test under CFL visible illumination was performed in 6 cycles (30-min UV exposure, 8 hours’ visible illumination, and 24 h in the darkness before every cycle) and showed that the CDs/TiO_2_@SAO composite coatings on wood materials had stable photostability after 6 cycles. These results fully implied that the as-prepared CDs/TiO_2_@SAO core-shell catalyst and the coated wood materials with CDs/TiO_2_@SAO composite coatings had stable photostability under ambient light sources and a dark environment. 

Figure 9 shows the afterglow self-luminous wood materials with CDs/TiO_2_@SAO composite coatings compared with the control wood. It was apparent that the composite coating was basically transparent and did not affect the color of the wood surface (Figure 9a), which is necessary for the indoor decorative wood materials. Moreover, as shown in Figure 9e–h, the wood materials with CDs/TiO_2_@SAO composite coatings presented long-term afterglow luminescence after suitable light energy storage (UV or visible light), which is consistent with the long-afterglow decay lifetime result of CDs/TiO_2_@SAO composites.

The potential photocatalytic mechanism for the CDs/TiO_2_@SAO self-luminous photocatalyst in the degradation of formaldehyde pollution was illustrated in Figure 10. For the pristine N,P-doped CDs, with UV light or visible light exposure, the electrons in the valentine band (VB) were partially excited to the conduction band (CB), and some of these excited electrons can react with O_2_ directly to produce ●O_2_^−^ radicals for the degradation of formaldehyde [42]. Additionally, the holes (h+) can attack the formaldehyde molecule and decompose it into CO_2_ and H_2_O ultimately. For the CDs/TiO_2_@SAO composite photocatalyst, with excitation of a light source (UV or visible light), electron transfers between tightly bonded N,P-doped CDs and TiO_2_ nanoparticles may occur and generate lots of hot electrons [43]. Meanwhile, the SAO long afterglow phosphor can capture and store luminous energy efficiently and give out visible light (green) in a dark environment. When the light excitation source was turned off, the CDs/TiO_2_@SAO composites could emit long-lasting luminescence (from strong to weak afterglow brightness) continually at night for a few hours, which could be absorbed and re-utilized by the CDs/TiO_2_ composite for photocatalytic degradation of formaldehyde [44,45]. Therefore, owing to the synergistic effect of CDs/TiO_2_ nanoparitcles and the afterglow luminescence from SAO phosphors, the as-prepared CDs/TiO_2_@SAO composite presents a continuous and efficient photocatalytic activity for the degradation of formaldehyde under indoor visible illumination and in a dark environment.

## 4. Conclusions

In conclusion, a novel efficient self-luminous wood coating with CDs/TiO_2_ nanomaterial coated long afterglow SAO composite (CDs/TiO_2_@SAO) was constructed for the photocatalytic degradation of indoor formaldehyde pollution. The photocatalytic degradation reaction of formaldehyde reached ~60% in a 160-min cycle under visible light (λ ≥ 400 nm) owing to the synergistic effect of potential electron transfer between fluorescent CDs and anatase TiO_2_ nanoparticles. In addition, owing to the long-lasting luminescence of SAO phosphor at night, the as-prepared CDs/TiO_2_@SAO composite photocatalyst can store luminous energy efficiently during the daytime so as to emit visible self-luminescence continuously for a few hours in the absence of light exposure. Furthermore, the wood coating with CDs/TiO_2_@SAO composites also maintained a continuous and efficient photocatalytic activity in ~360-min for degrading formaldehyde pollution in the darkness owing to the self-provided visible light from the SAO long-afterglow phosphors. The current research implies that fluorescent CDs and self-luminous SAO-modified traditional anatase TiO_2_ catalysts could enhance the photocatalytic activity and achieve long-term photocatalysis for indoor air clean-up in the daytime with an indoor visible light supply and a dark environment without any additional visible light sources.

## Figures and Tables

**Figure 1 polymers-15-02077-f001:**
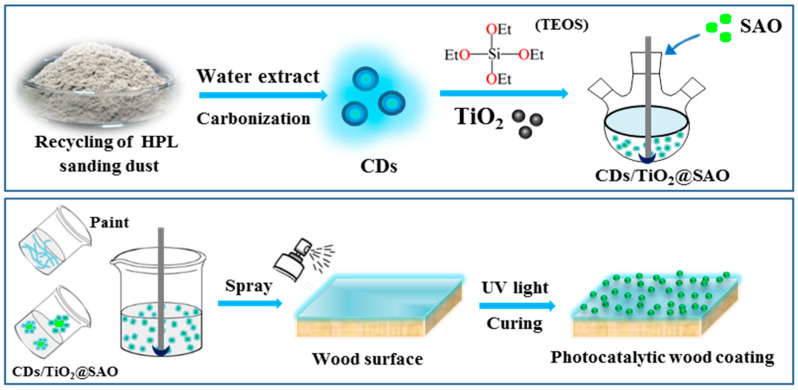
Preparation of the CDs/TiO_2_@SAO core-shell microspheres and the long-lasting photocatalytic wood coatings.

**Figure 2 polymers-15-02077-f002:**
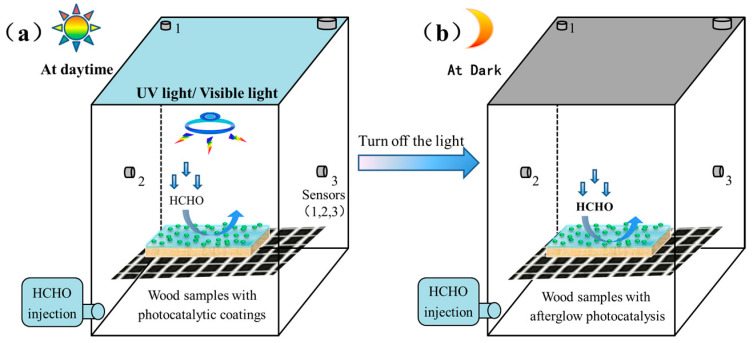
Schematic of the photocatalytic reactor for formaldehyde degradation: (**a**) photocatalysis under light source conditions (UV light or CFL visible light); (**b**) photocatalysis in the darkness.

**Figure 3 polymers-15-02077-f003:**
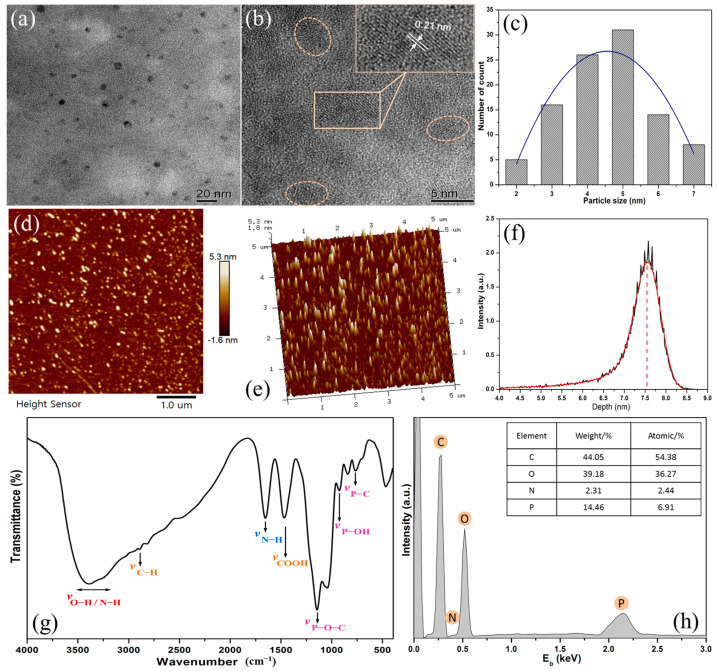
(**a**) TEM and (**b**) HRTEM images of N,P-doped CDs. Inset is the lattice fringes, the color circles are the magnification of CDs; (**c**) Histogram image of N,P-doped CDs (the color curve is the average particle size distribution); (**d**) 2D-AFM map, (**e**) 3D-AFM image, and (**f**) nanoparticle deposition depth of the N,P-doped CDs dispersion; (**g**) FTIR spectrum and (**h**) EDS diagram of the N,P-doped CDs.

**Figure 4 polymers-15-02077-f004:**
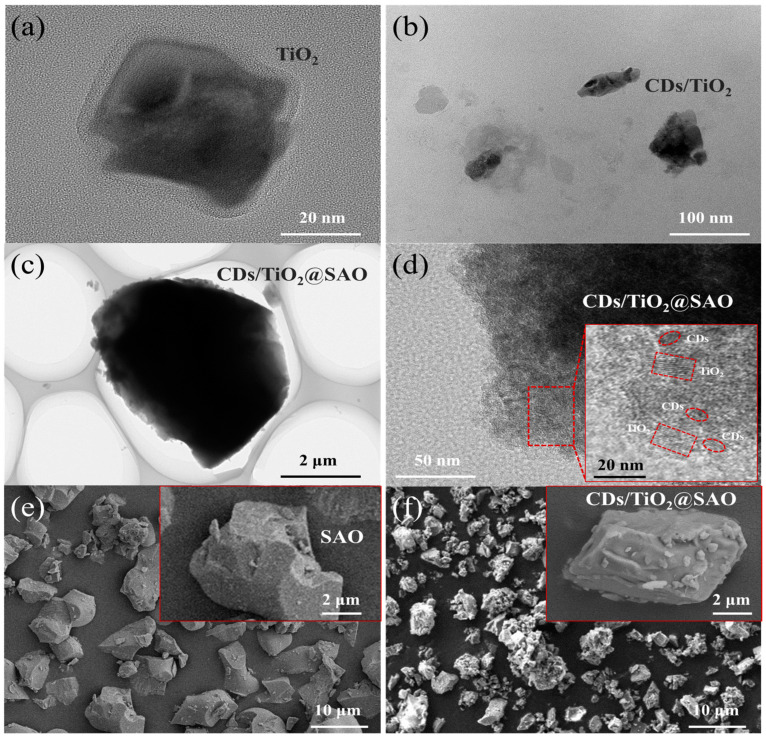
TEM images of (**a**) TiO_2_, (**b**) CDs/TiO_2_, and (**c**) CDs/TiO_2_@SAO composite photocatalyst. (**d**) HRTEM of CDs/TiO_2_@SAO. Inset is the lattice fringes of CDs/TiO_2_ coating on the SAO particle; SEM images of (**e**) uncoated SAO and (**f**) CDs/TiO_2_ modified SAO phosphors. Inset is the magnified image.

**Figure 5 polymers-15-02077-f005:**
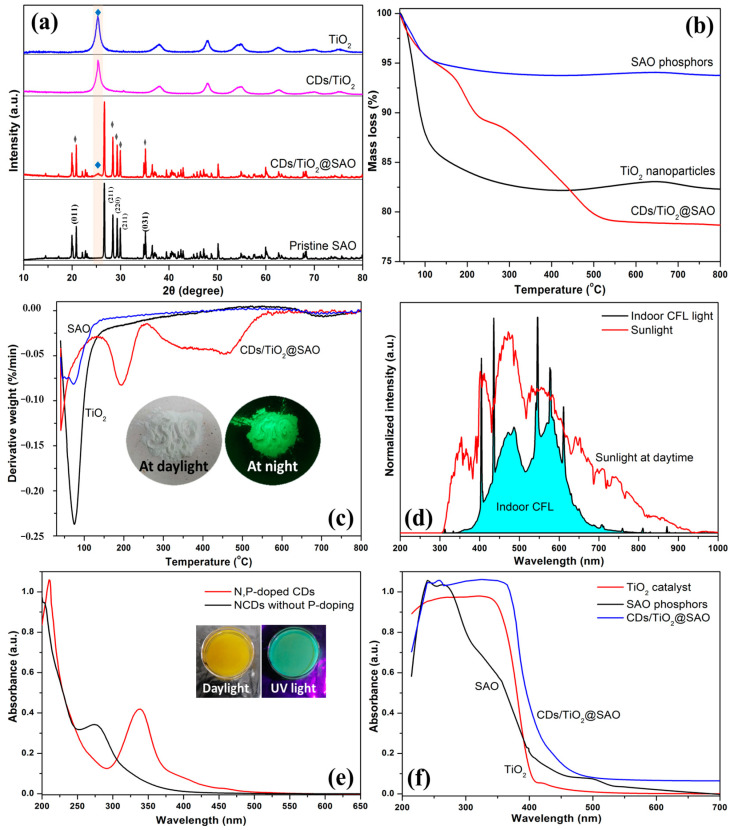
(**a**) XRD patterns of TiO_2_ and SAO phosphors with CDs coated modification; (**b**) TGA and (**c**) DTG of SAO, TiO_2_, and coated CDs/TO_2_@SAO particles. Inset is the photographs of CDs/TO_2_@SAO at daylight and dark condition after pyrolysis; (**d**) Emission spectra for the indoor CFL light and the sunlight at daytime; (**e**) UV-vis absorption spectra of different CDs. Inset was N,P-doped CDs solution under the daylight and 380 nm UV-lamp; (**f**) UV-vis diffuse reflectance spectra for different phosphors.

**Figure 6 polymers-15-02077-f006:**
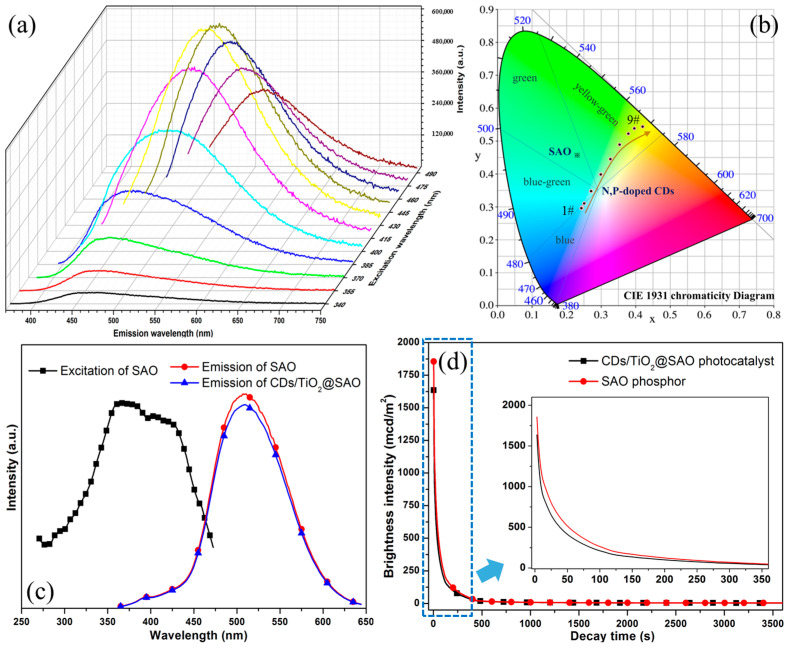
(**a**) PL emission spectra of N,P-doped CDs with λ_ex_ ranging in 340–490 nm (excitation gradient of 15 nm). The color curves are the emission spectra with different excitations; (**b**) CIE chromaticity diagram of SAO and N,P-doped CDs solution, where samples 1# through 9# correspond to PL emission with λ_ex_ ranging in 340–490 nm (excitation gradient of 15 nm); (**c**) Excitation spectrum of SAO and the PL emission of CDs/TiO_2_@SAO; (**d**) Long-afterglow decay lifetime of the CDs/TiO_2_@SAO under dark environment (detected after 365 nm irritation).

**Figure 7 polymers-15-02077-f007:**
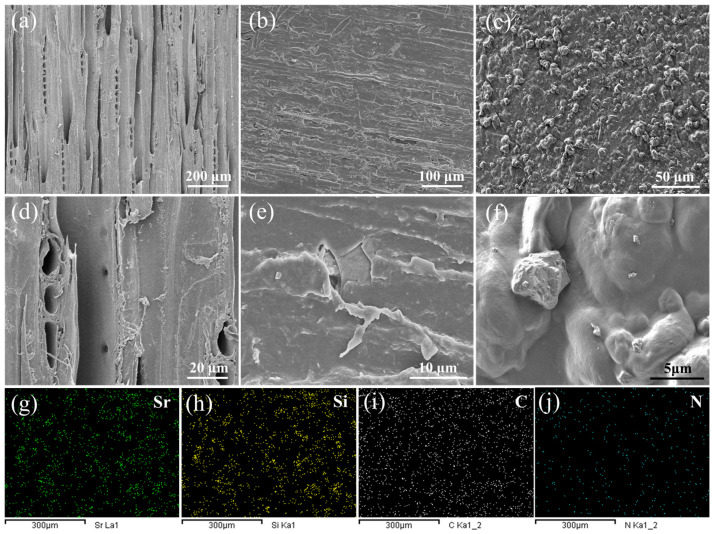
Morphologies and EDS scans of the coated wood surfaces: (**a**,**d**) Control wood; (**b**,**e**) Wood with pristine paint coatings; (**c**,**f**) Photocatalytic wood with CDs/TiO_2_@SAO composite coatings; (**g**–**j**) Element-mapping distributions of Sr, Si, C, and N, respectively.

**Figure 8 polymers-15-02077-f008:**
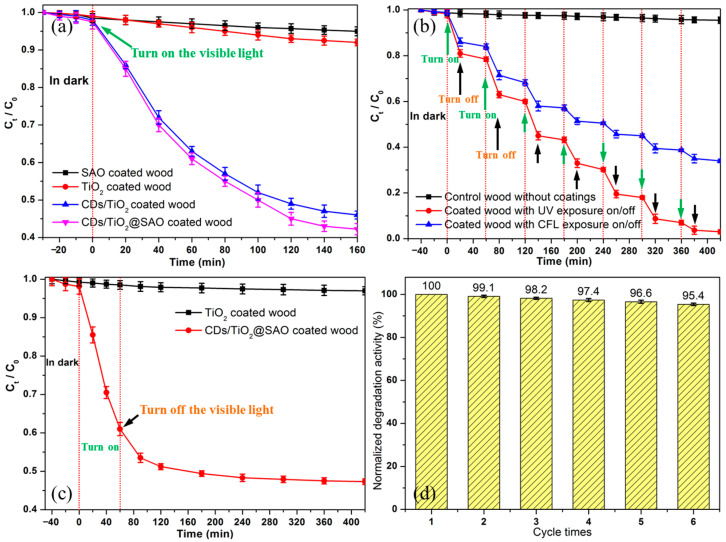
Photocatalytic activities of the CDs/TiO_2_@SAO composite coated wood materials for the degradation of formaldehyde with different irradiations: (**a**) Continuous CFL visible light (λ ≥ 400 nm) exposure; (**b**) Intermittent UV (λ = 365 nm) or CFL visible light exposure; (**c**) Long-lasting afterglow degradation in the darkness after the initial light exposure; (**d**) Recycling stability after different cycles.

**Figure 9 polymers-15-02077-f009:**
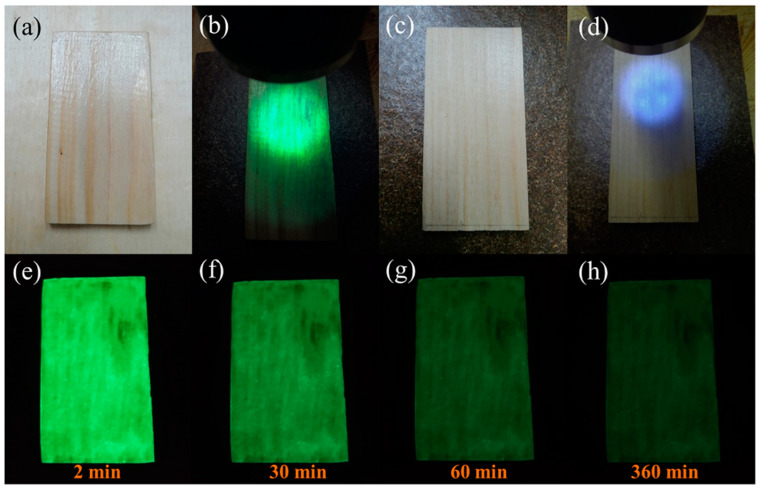
Long-lasting afterglow self-luminous wood materials with CDs/TiO_2_@SAO composite coatings compared with the control wood without luminescent coating: Functional wood with CDs/TiO_2_@SAO composite coatings (**a**) under daytime and (**b**) UV light exposure; Control wood (**c**) under daytime and (**d**) UV light exposure; (**e**–**h**) Self-luminous behavior of CDs/TiO_2_@SAO composite coatings at different lifetimes after the UV light exposure.

**Figure 10 polymers-15-02077-f010:**
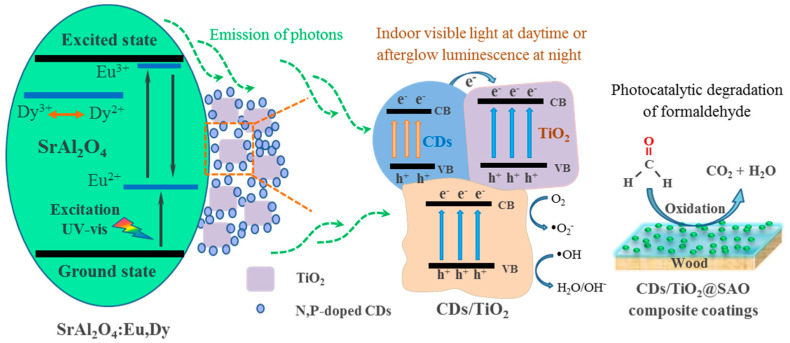
Schematic of the photocatalytic mechanism for the functional wood materials with the long-afterglow self-luminous CDs/TiO_2_@SAO composite photocatalyst in the degradation of formaldehyde pollution.

## Data Availability

The data that allowed for the writing of this article are available for request from the Wood Functional Materials department/Research Institute of Wood Industry/Chinese Academy of Forestry.

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
