# Peer review of "Self-Luminous Wood Coatings with Carbon Dots/TiO2 Grafted Afterglow SrAl2O4: Eu, Dy Core-Shell Phosphors for Long-Lasting Formaldehyde Removal"

_polymers, 2023, doi:10.3390/polym15092077_

Round 1

Reviewer 1 Report

Journal: Polymers (ISSN 2073-4360)

Manuscript ID: polymers-2304831

Type: Article

Title: Self-luminescent wood coatings with carbon dots/ TiO2 grafted afterglow SrAl2O4: Eu, Dy core-shell phosphors for long-lasting formaldehyde removal.

Authors: Longfei Zhang , Ying Wang , Limin Peng , Zhilin Chen , Shaoyi Lyu * , Siqun Wang.

Very good article

a)           Please; write the objective of the present work clearly.

b)          Why the author didn’t measure the optical (T, R, n K, Eg), electrical and mechanical properties of the samples?

c)           Why the author didn’t measure the dielectric properties of samples?

d)          It is better for the author to put images of the samples prepared?

e)           For references, choose recent refs. Please, refer to these refs for mechanical properties.

DOI: https://doi.org/10.1016/j.optmat.2022.112725

DOI:  https://doi.org/10.1016/j.mseb.2021.115191

Best Regards

Author Response

(1) Please; write the objective of the present work clearly. Agree. We have revised the sentences of the objective in the parts of Introduction and Conclusion to make it easier for readers to understand this article. (2) Why the author didn’t measure the optical (T, R, n K, Eg), electrical and mechanical properties of the samples? Why the author didn’t measure the dielectric properties of samples? In this study, we mainly modified the afterglow phosphors with CDs and TiO2 for the photocatalysis of indoor formaldehyde. As explained in the results, this core-shell material had good performance in degrading indoor formaldehyde, and the potential mechanism was also presented. The further mechanism for this phosphor including optical and electrical properties are being studied in another study for better understanding the long-lasting photocatalysis phenomenon. (3) It is better for the author to put images of the samples prepared? Agree. We have added some images of the samples prepared in Fig. 9. Moreover, the control images of the wood sample without photocatalysis coating were also provided so that the readers can better understand the wood functional product. (4) For references, choose recent refs. Please, refer to these refs for mechanical properties. DOI: https://doi.org/10.1016/j.optmat.2022.112725 DOI: https://doi.org/10.1016/j.mseb.2021.115191 Thanks. We have read these references, and refer to the suitable ref. in this study. Other revises by authors 4#: 1) We have added a few important references and changed the number of references accordingly. 2) We have also revised some grammar errors and phrases according to the reviewers’ comments.

Reviewer 2 Report

Referee report on manuscript “Self-luminescent wood coatings with carbon dots/ TiO2 grafted afterglow SrAl2O4: Eu, Dy core-shell phosphors for long-lasting formaldehyde removal

The article undoubtedly contains some new results that may be recommended for publication, but only after improvement and concretization of some incomprehensible points.

1. "Self-Luminescent" is found only in the title of the article. Readers remains to guess, and what is meant?

2.  SrAl2O4 doped with luminescent ions is the main object of this paper. Therefore, more attention in the Introduction  should be paid to this compound, its properties and applications. Moreover, taking into account how this material is prepared, point defects are necessarily created in it.  Although in SAO, point defects are almost not studied, nevertheless, they are studied in detail in close materials

See, for example, few recent reports:

Luchechko, A., Zhydachevskyy, Y., Ubizskii, S. et al. Afterglow, TL and OSL properties of Mn2+-doped ZnGa2O4 phosphor. Sci Rep 9, 9544 (2019). https://doi.org/10.1038/s41598-019-45869-7

Seeman, V., Feldbach, E., Kärner, T.,  et al (2019). Fast-neutron-induced and as-grown structural defects in magnesium aluminate spinel crystals with different stoichiometry. Optical Materials91, 42-49. https://doi.org/10.1016/j.optmat.2019.03.008

3. Line 53-60. What structure TIO2 is talking about?

4. How are the SAO samples of stichiometric?

5. What is the peak interpretration at 500 cm-1 (Fig.3 g).

6. Line 345, Please give more details on “to the oxygen-related P-doping groups”

7. Fig.5f. What is the reason for the step in the absorption at 350 nm for SAO?

7. Why is such a strong overlap between excitation and luminescent spectra (Fig.6).

What is in this case the band gap energy Eg value of SrAl2O4?  Compare with Table 1.1 in   paper: M.G. Brik, C.-G. Ma, T. Yamamoto, M. Piasecki, A.I. Popov. “First-principles methods as a powerful tool for fundamental and applied research in the field of optical materials. - Phosphor Handbook: Experimental Methods for Phosphor Evaluation and Characterization (CRC Press, Boca Raton), 2022, pp. 1-25.

https://doi.org/10.1201/9781003098669-1

8. The quality of many drawings is not satisfactory, it is difficult to see small details.

Author Response

The article undoubtedly contains some new results that may be recommended for publication, but only after improvement and concretization of some incomprehensible points.

Thank you for your time and patient review.

(1) "Self-Luminescent" is found only in the title of the article. Readers remains to guess, and what is meant?

Agree. We have supplied this definition in the introduction part and revised the expression of “Self-luminous” in the whole manuscript.

(2) SrAl2O4 doped with luminescent ions is the main object of this paper. Therefore, more attention in the Introduction should be paid to this compound, its properties and applications. Moreover, taking into account how this material is prepared, point defects are necessarily created in it. Although in SAO, point defects are almost not studied, nevertheless, they are studied in detail in close materials.

See, for example, few recent reports: Luchechko, A., Zhydachevskyy, Y., Ubizskii, S. et al. Afterglow, TL and OSL properties of Mn2+-doped ZnGa2O4 phosphor. Sci Rep 9, 9544 (2019). https://doi.org/10.1038/s41598-019-45869-7 Seeman, V., Feldbach, E., Kärner, T., et al (2019). Fast-neutron-induced and as-grown structural defects in magnesium aluminate spinel crystals with different stoichiometry. Optical Materials, 91, 42-49. https://doi.org/10.1016/j.optmat.2019.03.008 Agree. We have added some background information in the introduction part. Of course, we referred the ref. concerning the afterglow phosphor and its composites.

(3) Line 53-60. What structure TIO2 is talking about?

Yes, the structure of TiO2 in this study was anatase phase, we have re-emphasized the type of the TiO2 photocatalyst. In addition, we have also revised the expression in the part of Materials (2.1) and the entire manuscript.

(4) How are the SAO samples of stichiometric?

What is the peak interpretration at 500 cm-1 (Fig.3 g). According to our previous papers and the SEM image in Fig. 4e, the particle size of the SAO phosphors was irregular and distributed in a wide range owing to the fact that the SAO is a typical industrial product prepared by grinding method. Therefore, in the part of Materials of this manuscript, we provide the average particle size of SAO phosphors. As the FTIR peak of CDs at 500 cm-1 (Fig. 3 g), it was mainly ascribed to the complicated components in the recycling sanding dust.

(5) Line 345, Please give more details on “to the oxygen-related P-doping groups”

Agree. In Fig. 5e, we have presented the difference of the CDs in the presence and absence of P-doping, it was clear that the UV-vis spectra of both the CDs were obviously different. Considering the similar experimental condition except the P-doping process, we can obtain the conclusion that the additional visible light absorption mainly ascribed to the P-doping groups and surface defects in the N,P-doped CDs. Furthermore, we have added a ref. to support the conclusion in revision.

(6) Fig.5f. What is the reason for the step in the absorption at 350 nm for SAO?

Yes, as shown in Fig. 5f, the SAO possessed a strong diffuse reflectance at 200~400 nm, the slight increase at 350 nm mainly ascribed the structure of SAO and the uneven particle size. It also proved that the SAO had relatively strong absorption of UV (350 nm) light, which is benefit for the storage of light at daytime and emit visible light at dark environment.

(7) Why is such a strong overlap between excitation and luminescent spectra (Fig.6).

What is in this case the band gap energy Eg value of SrAl2O4? Compare with Table 1.1 in paper: M.G. Brik, C.-G. Ma, T. Yamamoto, M. Piasecki, A.I. Popov. “First-principles methods as a powerful tool for fundamental and applied research in the field of optical materials. - Phosphor Handbook: Experimental Methods for Phosphor Evaluation and Characterization (CRC Press, Boca Raton), 2022, pp. 1-25. https://doi.org/10.1201/9781003098669-1

Yes, we have provided the experimental condition of the emission spectra. The luminescent emission spectra were tested under 350 nm excitation rather than other strong absorption at 350~450 nm. Therefore, there was not obvious overlap between the excitation and luminescent spectra. The detailed mechanism with band gap energy will present in another study of our group. Moreover, we referred the above ref. concerning the energy expression of phosphor.

(8) The quality of many drawings is not satisfactory, it is difficult to see small details.

Yes, the pictures in the original manuscript (Word file) were clear with high resolution. Please check the Word file rather than the PDF in the system. Thanks for your detailed review. Other revises by authors 4#: 1) We have added a few important references and changed the number of references accordingly. 2) We have also revised some grammar errors and phrases according to the reviewers’ comments.

Reviewer 3 Report

This manuscript is very interesting. Zhang et al has concluded that efficient self-luminous wood coatings with CDs/TiO2 nanomaterial coated long afterglow SAO composite (CDs/TiO2@SAO) constructed for photocatalytic degradation of indoor formaldehyde pollution has  degradatded reaction of formaldehyde reached to ~60% in a 160-min cycle under visible light (λ≥ 400 525 nm). I would recommend to accept the work after few changes. Authors must provide high quality images and drawings. They are not of good quality. Is it possible to scale up the initial materials as coating? Schematic mechanism is not clear, how is the recombination mechanism is taking place? Proof of concept of the afterglow is not very clearly mentioned.  Why is afterglow important here in this work?  Please cite few relevant examples in introduction. 

Author Response

Comments and Suggestions for Authors This manuscript is very interesting. Zhang et al has concluded that efficient self-luminous wood coatings with CDs/TiO2 nanomaterial coated long afterglow SAO composite (CDs/TiO2@SAO) constructed for photocatalytic degradation of indoor formaldehyde pollution has degradatded reaction of formaldehyde reached to ~60% in a 160-min cycle under visible light (λ≥ 400 525 nm). I would recommend to accept the work after few changes. Thank you for your time and patient review. (1) Authors must provide high quality images and drawings. They are not of good quality. The images in the original manuscript (Word file) were clear with high resolution. Please check the Word file rather than the PDF in the system. Thanks for your detailed review. (2) Is it possible to scale up the initial materials as coating? Agree. We have added some images of the samples prepared. Moreover, the control images of the wood sample without photocatalysis coating were also provided so that the readers can better understanding the wood functional product. (3) Schematic mechanism is not clear, how is the recombination mechanism is taking place? Yes, we have revised the expression of the potential mechanism, the detailed mechanism for the CDs/TiO2/SAO system was further studied with an emphasis in the analysis of electronic energy bands. (4) Proof of concept of the afterglow is not very clearly mentioned. Why is afterglow important here in this work? Please cite few relevant examples in introduction. Yes, we have added some background information for the afterglow phosphors, and emphasized the importance of afterglow luminescence in the photocatalytic system, especially in the dark environment owing to their abilities to storage energy at daytime and further emit visible light at dark environment. Other revises by authors 4#: 1) We have added a few important references and changed the number of references accordingly. 2) We have also revised some grammar errors and phrases according to the reviewers’ comments.

Round 2

Reviewer 2 Report

The authors have successfully improved the original version of their manuscript, responding constructively to all the comments/recommendations of the reviewer.  Therefore, the article can be recommended for publication.